# Polyphonic Sound Event Detection Using Temporal-Frequency Attention and Feature Space Attention

**DOI:** 10.3390/s22186818

**Published:** 2022-09-09

**Authors:** Ye Jin, Mei Wang, Liyan Luo, Dinghao Zhao, Zhanqi Liu

**Affiliations:** 1Ministry of Education Key Laboratory of Cognitive Radio and Information Processing, Guilin 541006, China; 2School of Information and Communication, Guilin University of Electronic Technology, Guilin 541006, China; 3School of Information Science & Engineering, Guilin University of Technology, Guilin 541006, China

**Keywords:** sound event detection, temporal-frequency attention, feature space attention, convolutional recurrent neural networks, feature aggregation

## Abstract

The complexity of polyphonic sounds imposes numerous challenges on their classification. Especially in real life, polyphonic sound events have discontinuity and unstable time-frequency variations. Traditional single acoustic features cannot characterize the key feature information of the polyphonic sound event, and this deficiency results in poor model classification performance. In this paper, we propose a convolutional recurrent neural network model based on the temporal-frequency (TF) attention mechanism and feature space (FS) attention mechanism (TFFS-CRNN). The TFFS-CRNN model aggregates Log-Mel spectrograms and MFCCs feature as inputs, which contains the TF-attention module, the convolutional recurrent neural network (CRNN) module, the FS-attention module and the bidirectional gated recurrent unit (BGRU) module. In polyphonic sound events detection (SED), the TF-attention module can capture the critical temporal–frequency features more capably. The FS-attention module assigns different dynamically learnable weights to different dimensions of features. The TFFS-CRNN model improves the characterization of features for key feature information in polyphonic SED. By using two attention modules, the model can focus on semantically relevant time frames, key frequency bands, and important feature spaces. Finally, the BGRU module learns contextual information. The experiments were conducted on the DCASE 2016 Task3 dataset and the DCASE 2017 Task3 dataset. Experimental results show that the F1-score of the TFFS-CRNN model improved 12.4% and 25.2% compared with winning system models in DCASE challenge; the ER is reduced by 0.41 and 0.37 as well. The proposed TFFS-CRNN model algorithm has better classification performance and lower ER in polyphonic SED.

## 1. Introduction

Polyphonic sound event detection (SED) has attracted increasing research attention and numerous challenges [1,2] in recent years, and is mainly used for acoustic event classification and time detection. In real environments, multiple audio events may occur simultaneously. Polyphonic SED has important practical application value and theoretical significance [3]. In recent years, polyphonic SED has been applied to smart city traffic systems [4], equipment failure monitoring [5], smart home devices [6,7], telemedicine [8] and wildlife monitoring [9]. Environmental polyphonic sounds don’t show regular temporal patterns, such as phonemes in speech or rhythm in music. Therefore, it is difficult to accurately capture the characteristic information of time frames. In addition, the complexity and variability of polyphonic sound events in real scenes make them more difficult to detect.

To solve the above problems, various methods have been applied to perform polyphonic SED. In all polyphonic SED models, the extraction of acoustic features with strong characterization ability and effective classification algorithms are the keys to improving the overall model classification performance. The main features include linear predictive coding (LPC) [3], linear predictive cepstral coefficients (LPCC), discrete wavelet transform (DWT), Mel frequency cepstral coefficients (MFCCs) [3] and Log-Mel spectrograms (Log-Mel). Traditional classifiers include support vector machines (SVMs) [10], Gaussian mixture models (GMM) [11], hidden Markov models (HMM) [12], multi-layer perceptron (MLP) [13], and so on. However, these traditional models are only applicable to single acoustic events and small datasets [14]. With the increasing of dataset scale and audio complexity, the above traditional classification models cannot meet the classification requirements of the system. With the development of machine learning, classification models of the neural network are far superior to traditional classifiers, including feedforward neural networks (FNN), recurrent neural networks (RNN) [15], convolutional neural networks (CNN) [16] and convolutional recurrent neural networks (CRNN) [17,18,19,20]. In recent years, most participants of the DCASE challenge have used classification models based on deep learning for SED [1]. CRNN not only has the powerful ability of CNN to capture time-frequency features and process multi-dimensional feature information [21,22,23,24] but also has the advantages of RNN for sequence recognition. Therefore, CRNN is very suitable for polyphonic SED tasks [24]. In particular, RNN with the BGRU module provides better access to contextual information [25], which more accurately predicts the start times and offset times of each sound event.

In recent years, machine learning models based on attention mechanisms have been popularly adopted in image recognition [26], machine translation [27], text classification [28] and speech recognition [29,30]. In addition, relevant literature [31,32] has proved that networks based on attention mechanisms can further improve the classification performance of SED, such as channel attention [33], spatial attention [34,35], temporal attention [36] and frequency attention [37]. Hu et al. demonstrated the superiority of the channel attention mechanism [38]. Shen et al. presented a TF-attention mechanism for SED [37]. Li et al. focused on the mechanism of temporal attention by calculating the weight of the spectrograms [39]. The above studies of attention mechanisms just focused on the difference in time steps but ignored the importance of different frequency bands and different feature dimensions. Although classification models based on the neural network have been popularly applied in the acoustics field, the following challenges exist in the detection of environmental polyphonic sounds: (1) The temporal-frequency structure of polyphonic sounds is very complicated, which may be continuous (rain), abrupt (gunshots) or periodic (clock tick). (2) Discontinuity and uncertain duration of polyphonic sounds affect model classification performance. (3) The polyphonic SED model has larger datasets, more parameters and more feature space dimensions.

In polyphonic SED, this paper proposes an innovative model named TFFS-CRNN to address the mentioned challenges effectively. The TFFS-CRNN model contains four modules: dual-input TF-attention module, CNN module, FS-attention module and RNN with BGRU module. Our model innovatively combines the dual-input TF-attention mechanism and FS-attention mechanism, filtering out unimportant frequency bands information and weighting the important feature dimensions to characterize the key feature information of the polyphonic sound event. The TFFS-CRNN model can capture key temporal-frequency features and spatial features of sound events. In the experiment, the performance of the TFFS-CRNN model was evaluated on the public DCASE 2016 Task3 dataset and DCASE 2017 Task3 dataset. The TFFS-CRNN model proposed in this paper is compared with the DCASE challenge’s winning system models, which fully demonstrates the superiority of the TFFS-CRNN model in polyphonic SED. Our contributions are as follows: (1) The dual-input TF-attention mechanism was introduced for polyphonic sound events with complex time-frequency structures. (2) In the CRNN module, the multi-dimensional higher-order features are extracted to solve the problem of discontinuous polyphonic sounds. (3) In order to solve the problem of feature dimension redundancy, we introduce the FS-attention mechanism to weight important dimensions.

The rest of the paper is arranged as follows: Section 2 involves a detailed description of the TFFS-CRNN model and its rationale. Section 3 reports and compares the experimental results. Section 4 discusses our findings. Conclusions are placed in Section 5.

## 2. Methods

The overall structure of the proposed TFFS-CRNN model is shown in Figure 1, which includes five main components: the dual-input TF-attention module, the CNN module, the FS-attention module, the RNN module and the fully connected (FC) output layer. Firstly, the MFCCs feature and Log-Mel spectrograms are extracted as input of the dual-input TF-attention module and integrated into the TF-attention spectrograms of two features. The dual-input TF-attention features not only enhance the representation capabilities of features for polyphonic sounds, but also highlight important time-frame information and key frequency-bands information. CNN is responsible for extracting multi-dimensional higher-order features from the TF-attention feature of MFCCs and Log-Mel spectrograms. The FS-attention module dynamically learns the importance of each dimension of multi-dimensional high-order features and gives different attention to different dimensions. The FS-attention module could extract important feature space information and ignore unimportant dimensions. The RNN module uses 32-unit BGRU to gain contextual information and predict the start times and offset times of polyphonic sounds accurately. Finally, the output features of BGRU are fed into the FC layer to obtain the classification results of the TFFS-CRNN model. In this section, the whole network model is described in detail.

### 2.1. Dual-Input Temporal-Frequency Attention

This sub-section focuses on the dual-input TF-attention algorithm model shown in Figure 2. The polyphonic SED model introduces a TF-attention mechanism that assigns larger weight values to the critical time frames and critical frequency bands. At the same time, the attention weight of time frames and irrelevant bands with less information will be decreased. Log-Mel features *X_Log-Mel_* ∈ *R^F^*^×*T*^ and MFCCs features *X_MFCCs_* ∈ *R^F^*^×*T*^ are chosen as inputs to the dual-input TF-attention model. The harmonic spectrograms of the two features can visually and clearly display the frequency band information of polyphonic sounds, and the vertical structure of the percussive spectrograms can highlight the difference between time frames and noise. Therefore, the TF-attention algorithm model uses the harmonic-percussive source separation (HPSS) algorithm [32]. Then, the harmonic spectrograms and percussive spectrograms of Log-Mel and MFCCs features will be gained via HPSS. The harmonic spectrograms are input to the frequency attention network (F-Attention-Net) and the percussive spectrograms are fed into the temporal attention network (T-Attention-Net). The specific structure is shown in Figure 2. First, the Log-Mel spectrograms *X_Log-Mel_* ∈ *R^F×T^* or the MFCCs *X_MFCCs_* ∈ *R^F×T^* will be normalized. After that, we perform the convolution operations on the harmonic and percussive spectrograms to extract high-order features. The convolution kernel sizes are (1 × 3) and (2 × 1), respectively. A couple of convolution operations reduce the time dimension of harmonic spectrograms to 1. Similarly, the same convolution operation is performed on the frequency dimension of percussive spectrograms. After that, we use the (1 × 1) channel convolution kernel to compress. In this way, we can obtain the *I_F_* ∈ *R^F×^*^1^ with size (F,1) and the *I_T_* ∈ *R*^1*×T*^ with size (1,T). Finally, we normalize *I_F_* ∈ *R^F×^*^1^ and *I_T_* ∈ *R*^1*×T*^ by using the SoftMax function, thus obtaining the frequency weight *F**_W_* ∈ *R^F×^*^1^ and the temporal weight *T**_W_* ∈ *R*^1*×T*^.
(1)$$F_{Logmel-w}(f)=\frac{\exp(I_{Logmel-F}(f,1))}{\sum_{i=1}^{F}\exp(I_{Logmel-F}(i,1))}$$
(2)$$F_{MFCCs-w}(f)=\frac{\exp(I_{MFCCs-F}(f,1))}{\sum_{i=1}^{F}\exp(I_{MFCCs-F}(i,1))}$$
(3)$$T_{Logmel-w}(t)=\frac{\exp(I_{Logmel-T}(1,t))}{\sum_{j=1}^{T}\exp(I_{Logmel-T}(1,j))}$$
(4)$$T_{MFCCs-w}(t)=\frac{\exp(I_{MFCCs-T}(1,t))}{\sum_{j=1}^{T}\exp(I_{MFCCs-T}(1,j))}$$
where 1 ≤ *t* ≤ *T* and 1 ≤ *f* ≤ *F* in Equations (1)–(8). Next, the spectrogram *X_Log-Mel_* ∈ *R^F×T^* or *X_MFCCs_* ∈ *R^F×T^* point-multiplied with the attention weight matrices along the temporal and frequency directions respectively to produce the frequency attention spectrogram *A_F_* ∈ *R^F×T^* and the temporal attention spectrogram *A_T_* ∈ *R^F×T^* with the following expressions.
(5)$$A_{Logmel-F}(f)=X_{Logmel}(F,t)*F_{Logmel-w}$$
(6)$$A_{MFCCs-F}(f)=X_{MFCCs}(F,t)*F_{MFCCs-w}$$
(7)$$A_{Logmel-T}(t)=X_{Logmel}(f,T)*T_{Logmel-w}$$
(8)$$A_{MFCCs-T}(t)=X_{MFCCs}(f,T)*T_{MFCCs-w}$$

As the time and frequency domains of spectrograms contain information about temporal and frequency features, respectively, they are quite different from images in the image classification domain. The TF-attention mechanism assigns different weights to the time frame and frequency band. Both temporal and frequency features can be enhanced by the TF-attention mechanism, improving the ability of features to characterize the key feature information of the polyphonic sound event. In general, the combined approach is available in either parallel or concatenation. However, the concatenation methods combine two attention mechanisms, leading to poor performance. In this paper, three fusion strategies are designed, including average combination, weighted combination and channel combination. In the experiment in Section 4.2, the weighted combination strategy has the best performance. So, the weighted combination strategy was chosen to merge the temporal attention mechanisms and frequency attention mechanisms into a unified model. In the training, set up two variable parameters *α* and *β* and *α* + *β* = 1. Finally, the TF-attention spectrogram *A_T_**_&F_* is computed as follows:(9)$$A_{Logmel-T\&F}=\alpha A_{Logmel-T}+\beta A_{Logmel-F}$$



(10)
$$A_{MFCCs-T\&F}=\alpha A_{MFCCs-T}+\beta A_{MFCCs-F}$$



Log-Mel-TF-attention and MFCCs-TF-attention fusion strategies: The first strategy is called average combination. The obtained Log-Mel-T&F-Attention and MFCCs-T&F-Attention are fused in a 1:1 ratio to produce the final temporal-frequency attention spectrogram *LM-T&F-Average*. The detailed process is as follows:(11)$$A_{LM-T\&F-A\mathrm{verage}}=A_{Logmel-T\&F}+A_{MFCCs-T\&F}$$

The second strategy is called weighted combination. In the training, set up two variable parameters *λ* and *μ* and *λ* + *μ* = 1. Then the L-TF-attention and M-TF-attention spectrograms are fused according to the ratio of the parameters. Finally, the TF-attention spectrogram *LM-T&F-Weight* that fuses the two features of Log-Mel and MFCCs is then obtained. The detailed process is as follows:(12)$$A_{LM-T\&F-W\mathrm{eight}}=\lambda A_{Logmel-T\&F}+\mu A_{MFCCs-T\&F}$$

The third strategy is called channel combination. The obtained two attention spectrograms L-TF-attention and M-TF-attention are concatenated as two-channel output. The process is as follows:(13)$$A_{LM-T\&F-C\mathrm{hannel}}=\mathrm{joint}(A_{Logmel-T\&F};A_{MFCCs-T\&F})$$

### 2.2. Feature Space Attention

This sub-section focuses on the FS-attention algorithm model shown in Figure 3. In the neural network model, all dimensions of the feature spectrogram are treated equally, thus drowning out some important feature dimensions that contain critical information. The TFFS-CRNN model uses the FS-attention module to obtain the important feature dimension. Each dimension of multi-dimensional high-order features was assigned different learnable weights to obtain the space attention features *F_Attention_* ∈ *R^T×K×F^*.

The feature space attention algorithm is shown in Figure 3, the multi-dimensional high-order features *F_LM_* ∈ *R^K×T×F^* are obtained from the CNN module, where *K* represents the number of channels, *F* represents the frequency and *T* represents the time frame length. Then the high-order feature *F_LM_* ∈ *R^K×T×F^* was input to the FS-attention model. The FS-attention contains a SoftMax activation layer and a FC feedforward layer to calculate the importance weight of each feature dimension of *F_LM_*. The importance weight *I* is the outputs of the SoftMax layer, which is assigned to different feature dimensions. First, *F_LM_* ∈ *R^K×T×F^* is permuted into a 3-dimensional tensor $F_{LM}^{\prime }$.
(14)$$F_{LM}^{\prime }=R^{K\times T\times F\to T\times K\times F}(F_{LM})$$
where *R^K×T×F^*^→^*^T×K×F^* denotes the *F_LM_* with dimension *K × T × F* becomes the $F_{LM}^{\prime }$ with dimension *T × K × F*. Subsequently, $F_{LM}^{\prime }$ is flattened as a 2-dimensional tensor $F_{LM}^{"}$ by fixing the time dimension *T*, as shown in the following:(15)$$F_{LM}^{"}=R^{T\times K\times F\to T\times KF}(F_{LM}^{\prime })$$

Next, the $F_{LM}^{"}$ is input to the feedforward layer. In the FC feedforward layer, the number of hidden units is set to *KF*. The dimension of weights *I* is *M* = *KF*, which is expressed as follows:(16)$$I=\left\{I_{1},I_{2},\;\cdots ,I_{d},\;\cdots ,I_{M}\right\}$$
where *I_m_* influences the *m*th dimensional feature of $F_{LM}^{"}$, the expression of *I_m_* is:(17)$$I_{m}=\frac{\exp(O_{m})}{\sum_{j=1}^{j=m}\exp(O_{j})}$$

The dimension of $F_{LM}^{"}$ is *M*. The *j*th dimensional output of the SoftMax activation layer is *O_j_*. The importance weight *I* is repeated *T* times, and the dimension of weight *I* becomes *T × K × F*. The FS-attention vector *I^’^* ∈ *R^T×K×F^* can be expressed as:(18)$$I^{\prime }=R^{T\times KF\to T\times K\times F}(I)$$
where *R^T×KF^*^→^*^T×K×F^* denotes the feature space importance weight *I* is reshaped from *T × KF* to *T × K × F*. The outputs of the FS-attention module can be expressed as:(19)$$F_{Attention}=I^{\prime }\circ F_{LM}^{\prime }$$
where “◦” represent the Hadamard product. Then the outputs *F_Attention_* of the FS-attention module are fed into the RNN with the BGRU module.

### 2.3. BGRU

In the RNN module, BGRU is used to learn contextual information from higher-order attention features, which can more accurately predict the start times and offset times of each sound event. As is shown in Figure 4, the BGRU module consists of two gated recurrent units (GRU), training direction of each neuron is reversed. Therefore, the correlation between the pre and post information can be fully exploited and information from both past and future directions can be introduced. GRU integrates the forget gate and the input gate into one gate called the update gate (denoted as *z_t_*), and the GRU unit only contains the update gate and the reset gate. The update gate is mainly used to control how much of the memory information from the *h*_*t*−1_ moment is retained at the current moment. While the reset gate determines how much of the memory information from the previous moment is combined with the new input information to form the new memory content. The single-layer GRU and the output vector *y* are computed as:(20)$$z_{t}=\sigma_{g}(W_{z}x_{t}+U_{z}h_{t-1}+b_{z})$$
(21)$$r_{t}=\sigma_{g}(W_{r}x_{t}+U_{r}h_{t-1}+b_{r})$$
(22)$${h^{\prime }}_{t}=\tanh(W_{h}x_{t}+U_{z}(r_{t}⊗h_{t-1})+b_{h})$$
(23)$$h_{t}=(1-z_{t})⊗h_{t-1}+z_{t}⊗{h^{\prime }}_{t}$$
(24)$$y=\sigma_{g}(W_{0}h_{t}+b_{0})$$
where *z_t_* is the output of the update gate, *r_t_* is the output value of the reset gate, *y* ∈ [0, 1]*^N^*, *N* represents the number of sound events. The symbol ⊗ denotes matrix multiplication. *W** and *U** in Equations (20)–(24) denote the weight matrix, *b** represents the bias term, and *δ_g_* represents the sigmoid activation function. Each dimension of *y_t_* means the probability of a certain sound event occurring at time *t*. If the hidden layer of GRU has a dimension of *d*, then *r_t_*, *z_t_*, *h_t_*, $h_{t}^{\prime }$ ∈ *R^d^*. Using the BGRU module further enhances the performance of the TFFS-CRNN model. The binary cross entropy loss function (BCE loss) of the BGRU is expressed as:(25)$$loss(y,\hat{y})=-\frac{1}{N}\sum_{i=1}^{N}\left[{\hat{y}}_{i}\log(y_{i})+(1-{\hat{y}}_{i})\log(1-y_{i}\right)]$$

*N* represents the number of sound events, and $\hat{y}\in {\left\{0,1\right\}}^{N}$ is the binary indicator of sound events.

## 3. Experiment

This section describes the experimental datasets, evaluation metrics and experimental configurations in the domain of polyphonic SED [40,41,42]. Experiments are conducted on publicly available datasets to validate the effectiveness of the model, and then the results of the method provided in this study are compared with the results of existing methods.

### 3.1. Environmental Sound Datasets

In this study, the DCASE 2016 Task3 dataset and DCASE 2017 Task3 dataset [42,43] were used to evaluate our proposed TFFS-CRNN model. It is the datasets of DCASE 2016 Task3 and DCASE 2017 Task3 mentioned above. The experimental results were compared with the winner systems. The datasets are as follows.

Both datasets contain environmental sounds of daily life, which are divided into indoor and outdoor scenes. The audio of the DCASE 2016 Task3 dataset is mono with 44.1 kHz sampling rate. The DCASE 2017 Task3 dataset contains more street sounds and human voices in real-life recordings. Each audio of the DCASE 2017 Task3 dataset is 3–5 min in length and the sampling frequency is also 44.1 kHz. As shown in Table 1, the DCASE 2017 Task3 includes two daily environments: an outdoor residential area and an indoor home. Both the DCASE 2016 Task3 dataset and DCASE 2017 Task3 dataset contain a development set accounting for 70% of the total sample, and an evaluation set accounting for 30%. In this study, the four-fold cross-validation method is used to train and test.

### 3.2. Evaluation Metrics

In this paper, we use the widely recognized measurements proposed for polyphonic in [40] to compare the performance of different algorithms. In the experiment of this paper, the evaluation metrics are the segment-based F1-score (*F*1) and the error rate (*ER*). In addition, *F*1 is the harmonic average of precision (*P*) and recall (*R*), which takes values in the range of 0–1. The following equation describes the calculation process:(26)$$P=\frac{\sum TP}{\sum TP+\sum FP},\;R=\frac{\sum TP}{\sum TP+\sum FN},\;F1=\frac{2P\cdot R}{P+R}$$
where *TP*, *FP*, and *FN* denote true positive, false positive, and false negative respectively. The *ER* is the number of samples classified incorrectly. The *ER* is calculated as follows:(27)$$ER=\frac{\sum_{t=1}^{T}S(t)+\sum_{t=1}^{T}I(t)+\sum_{t=1}^{T}D(t)}{\sum_{t=1}^{T}N(t)}$$
where *T* is the number of sound events in segments *t*. Substitution events *S*(*t*) indicates the number of events where the model misclassifies sound event A as sound event B. Insertion event *I*(t) refers to an event A that is only detected in the model output, but no type of event occurs in the tag annotation at this moment. Deleted events *D*(*t*) refers to sound events that were present but not detected. *N*(*t*) is the total number of acoustic events from the annotations.

### 3.3. Experimental Configurations

In this research, the TFFS-CRNN model algorithm was all accomplished in the Python language. In this paper, all audio datasets are 44.1 kHz mono wave files, and the dimension of Log-Mel spectrograms and MFCCs is 40 × 256 (T = 256, F = 40). The frame size is 40 ms and the overlapping frames are 50%. As shown in Table 2, the hyperparameters of the TFFS-CRNN model were enhanced with a random search strategy. The CRNN model architecture used the Adam optimizer to train, and the ReLU activation function was used to introduce non-linearity. The value of the learning rate was 0.0001. In this study, we used early stopping to solve the problem of overfitting. Then, batch normalization (BN) of the TFFS-CRNN model aims to reduce the insensitivity of the network to initialization weights, using a loss rate of 0.25 after each convolutional layer. The global threshold *τ* is set to 0.5. The *τ* is used to determine the active acoustic events. If above the threshold, we determine that a segment contains the event classes. Table 2 shows the specific parameter settings for the CRNN network from the input to the output.

## 4. Discussion

The performance of the TFFS-CRNN model proposed was evaluated under different fusion strategies, different features, different classifiers and different methods. Simultaneously, we designed the three experiments on the DCASE 2016 Task3 dataset and DCASE 2017 Task3 dataset.

### 4.1. Comparison of Different Features

Six of the most used features were chosen for the experiments, including the Log-Mel spectrograms, MFCCs and short-time Fourier transform (STFT). The sampling rate of audio signals was 44.1 kHz, the frame size was 40 ms and the frame overlap was 50%. The STFT is computed at 1024 points with a size of 40 × 256. The other three features were the fusion of Log-Mel spectrograms, MFCCs and STFT. *F_MS_* is the fusion of MFCCs and STFT. *F_LS_* is the fusion of Log-Mel spectrograms and STFT. *F_LM_* is the fusion of Log-Mel spectrograms and MFCCs. In the process of experiments, the same TFFS-CRNN algorithm was used for all features to compare the classification effect of different features. Table 3 and Figure 5 show the comparison results of different features on the TFFS-CRNN model.

Compared with the other features, *F_LM_* improved F1 and ER values and has a better feature characterization capability. In experiments on the DCASE 2016 Task3 dataset, the performance of the aggregation feature *F_LM_* reached a maximum F1 of 60.2% and ER of 0.40. Using the DCASE 2017 Task3 dataset, its F1 and ER were 66.9% and 0.49, respectively. The results of experiments indicate that the fusion of Log-Mel spectrograms and MFCCs enhanced the classification performance. In contrast, the performances of the individual feature were poorer than the aggregated features.

### 4.2. Comparison of Different Fusion Strategies

In the previous section, *F_LM_* is the best performance feature, so *F_LM_* is used as the input feature in the fusion strategy comparison experiment. As shown in Table 4, three fusion strategies are proposed, including averaging combination, weighted combination and channel combination. The principle is shown in Equations (11)–(13). The experimental result shows that the second fusion strategy is the best compared with others. The learnable weight factor is more adaptive. So, we choose the second strategy to fuse Log-Mel-T&F-Attention and MFCCs-T&F-Attention. In the same way, we experiment on the strategies to fusing Log-Mel-T-Attention and Log-Mel-F-Attention. The experiment results show the weighted combination strategy performed best. In this way, the performance of the TF-attention mechanism can be optimally tuned.

### 4.3. Comparison of Different Classifiers

Compared to the other different classifiers, the proposed model TFFS-CRNN improved the F1 and reduce the ER. The comparison experiments of this section were under the same situation, using the same feature *F_LM_.* Figure 5 shows the detailed parameters for the TFFS-CRNN network. Other classifiers include DNN, CNN, RNN, CRNN, FS-CRNN, TF-CRNN and TFFS-CRNN. The following are the primary parameter settings. SVM: One-vs-Rest SVMs, sigmoid function. CNN: five convolutional layers, ReLU activation function. RNN: two BGRU recurrent layers, a time-distributed fully connected layer; CRNN: three layers of CNN and an RNN with BGRU. FS-CRNN: an FS-attention layer, three convolutional layers and two BGRU recurrent layers. TF-CRNN: a TF-attention layer, three convolutional layers and two recurrent layers with BGRU. TFFS-CRNN: a TF-attention layer, three convolutional layers, an FS-attention layer and two recurrent layers with BGRU. Table 5 shows the comparison results of the different classifiers using the feature *F_LM_*.

As shown in Table 5 and Figure 6, on the DCASE 2016 Task3 dataset, the F1 and ER values of the FS-CRNN network improved by 4.5% and 18%, respectively, compared with the CRNN model; the F1 and ER of TF-CRNN improved by 15.4% and 29%, respectively, compared with the CRNN; the F1 and ER values of TFFS-CRNN increased the F1 and ER values by 18.7% and 15%, respectively, compared with the FS-CRNN; TFFS-CRNN increased the F1 and ER values by 7.8% and 4%, respectively, compared with the TF-CRNN, and TFFS-CRNN increased the F1 and ER values by 23.2% and 33%, respectively, compared with the CRNN.

On the DCASE 2016 Task3 dataset, the F1 and ER values of FS-CRNN improved by 3.3% and 7%, respectively, compared with the CRNN; the F1 and ER values of TF-CRNN improved by 13.5% and 15%, respectively, compared with the CRNN; the F1 and ER values of TFFS-CRNN improved by 18.6% and 9%, respectively, compared with the FS-CRNN; TFFS-CRNN improved by 5.1% and 1%, respectively, compared with TF-CRNN, and TFFS-CRNN improved by 21.9% and 16%, respectively, compared with CRNN.

The experiment demonstrates the effectiveness of the attention mechanism. Compared with the CRNN model, the TF-CRNN model and FS-CRNN model have improved performance to a certain extent. In particular, the TFFS model combines the advantages of the TF-attention attention mechanism and the FS-attention at the same time, and its performance is greatly improved. We can conclude that the extraction of key temporal-frequency feature information and key dimensions information can greatly improve the performance of the TFFS-CRNN model in polyphonic SED. The improvement may be due to the increased attention to key information and the enhancement of the feature representation ability. Other classifiers were only a little improved, because of the submergence of key temporal-frequency information and key feature dimensions.

### 4.4. Comparison of Different Methods

The model provided was then compared with existing methods. Other compared models are specified below:

MFCCs+CRNN [43]: the model uses CRNN as the classifier and MFCCs feature as input.

MFCCs+GMM [42]: the model uses GMM as the classifier and MFCCs feature as input, which is the baseline model for DCASE2016 task3,

MFCCs+CNN [44]: the network model is a three-layer CNN, and the input feature is MFCCs.

Log-Mel+CaspNet [45]: the model uses Capsule Neural Networks (CaspNet), and the input feature is Log-Mel spectrograms.

Log-Mel+CRNN [46]: the network model is CRNN and the Log-Mel spectrograms as input.

Log-Mel+RNN [15]: the network model uses bidirectional LSTM RNN as the classifier, and the input feature is Log-Mel spectrograms.

Log-Mel+CNN [16]: the network model consists of two convolutional layers and two FC layers, and the input feature is Log-Mel spectrograms.

Log-Mel-CRNN [17]: the network model is CRNN and the Log-Mel spectrograms as input.

Binaural Mel energy + CRNN(BGRU) [47]: the model uses a capsule network with pixel-based attention and BGRU for polyphonic SED tasks, and the input feature is Binaural Mel energy.

Log-Mel+CRNN [48]: the network model is CRNN with GRU, and the input feature is Log-Mel spectrograms.

The experimental results in Table 6 and Figure 7 show that the proposed TFFS-CRNN model outperforms other methods. On the two TUT sound event datasets, F_LM_ + TFFS-CRNN achieved the best (F1, ER) performance with F1 scores of 60.2% and 66.9% and ERs of 0.40 and 0.42, respectively. Compared with the winning systems of DCASE challenge, the F1 improve 12.4% and 25.2%, and the ER is reduced 0.41 and 0.37 as well. Compared with the baseline systems of DCASE challenge, the F1 improve 25.9% and 24.1%, and the ER is reduced 0.44 and 0.52 as well. Compared to the latest algorithmic models, the TFFS-CRNN model still has superior performance.

The feature fusion strategy adds rich feature information, and the TF-attention mechanism and FS-attention mechanism increase the weight of key feature information, thus improving the feature representation ability. The experiments fully demonstrate that the combination of TF-attention and FS-attention can greatly improve the performance in the polyphonic SED, as the network can focus on the key information of the feature. Other methods without attention mechanisms drown out critical temporal-frequency information and key feature dimensions, and the characterization capability of a single feature is weaker than the fusion feature.

## 5. Conclusions

In this study, the TFFS-CRNN model is proposed for polyphonic SED. By introducing the TF-attention mechanism and FS-attention mechanism into the basic CRNN architecture, the TF-attention mechanism used for representational learning can effectively capture the key temporal-frequency features, and FS-attention can effectively enhance the features of important dimensions. The experiments of DCASE 2016 Task3 dataset and DCASE 2017 Task3 dataset show that the F-score is improved to 60.2% and 66.9%, and the ER is reduced to 0.40 and 0.42, respectively. The TFFS-CRNN model has a better classification performance than previous models for polyphonic SED. In particular, the model is far superior to the baseline systems and the winning systems of the DCASE challenge. In addition, this paper uses the BGRU module and FC layer to collect the previous moment state and the future moment state, thus obtaining contextual information. To further improve the generalization capability of the TFFS-CRNN model in polyphonic SED tasks, this paper also needs to enhance the training on weakly labeled datasets, and semi-supervised or unsupervised models are still worthy of further investigation in SED.

## Figures and Tables

**Figure 1 sensors-22-06818-f001:**
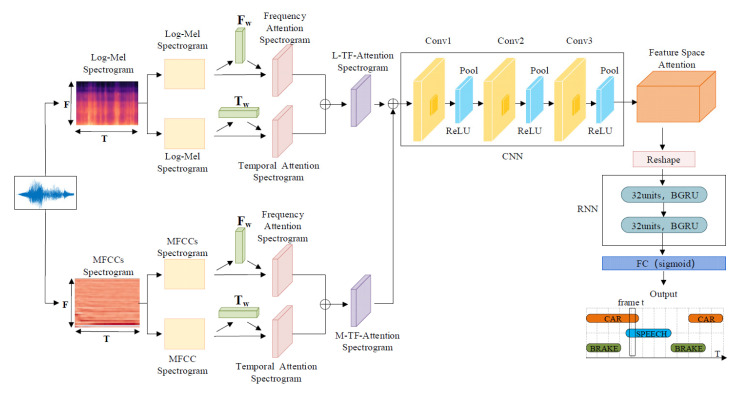
TFFS-CRNN network structure model.

**Figure 2 sensors-22-06818-f002:**
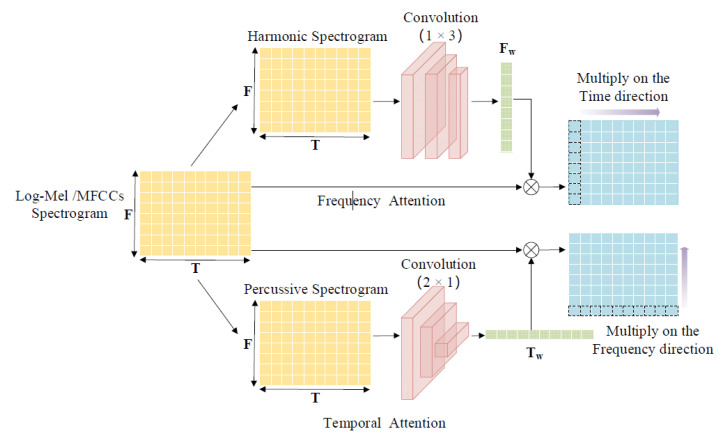
Temporal-frequency attention mechanism.

**Figure 3 sensors-22-06818-f003:**
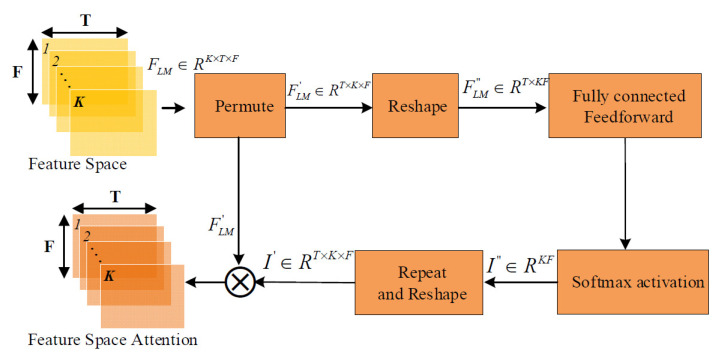
Feature space attention mechanism.

**Figure 4 sensors-22-06818-f004:**
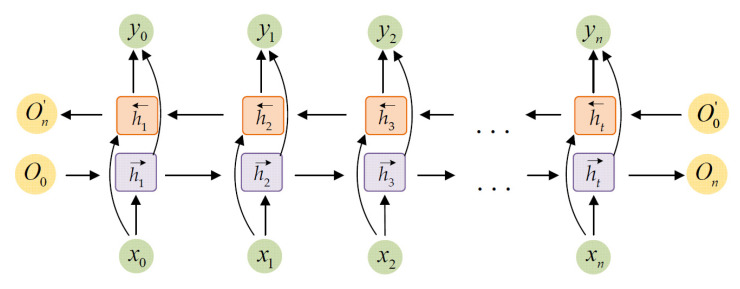
The BGRU model includes the forward GRU and the backward GRU. The input sequence is *X* = [*x*_0_, *x*_1_, *…*, *x_n_*]. The BGRU model is computed from *x*_0_ forward and then from *x_n_* backward. The final output is the combination of the forward output and backward output.

**Figure 5 sensors-22-06818-f005:**
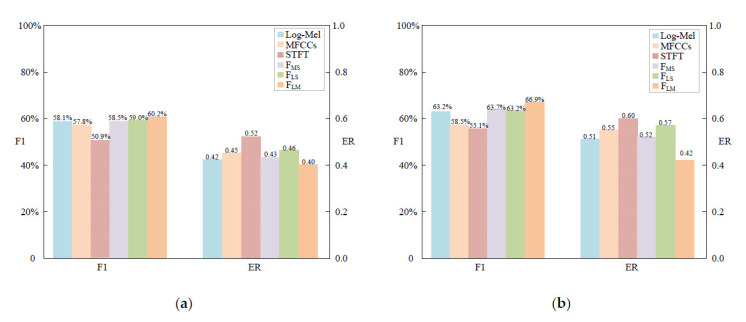
The results of different features on TFFS-CRNN in the evaluation dataset. The vertical coordinate of each bar in different colors represents ER and F1 of the model, respectively. The figure (**a**) is the experimental result of the DCASE 2016 Task3 dataset; the figure (**b**) is the experimental result of the DCASE 2017 Task3 dataset.

**Figure 6 sensors-22-06818-f006:**
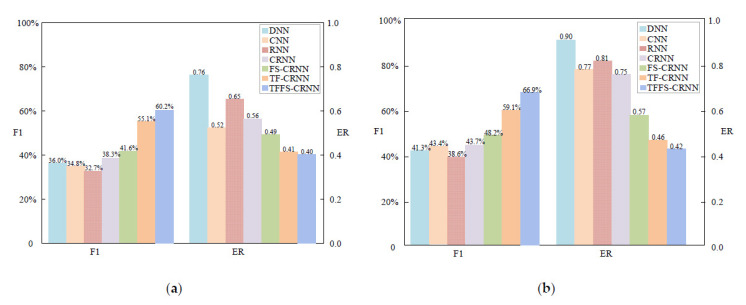
The results of different classifiers in the evaluation dataset. The figure (**a**) is the experimental result of the DCASE 2016 Task3 dataset; the figure (**b**) is the experimental result of the DCASE 2017 Task3 dataset.

**Figure 7 sensors-22-06818-f007:**
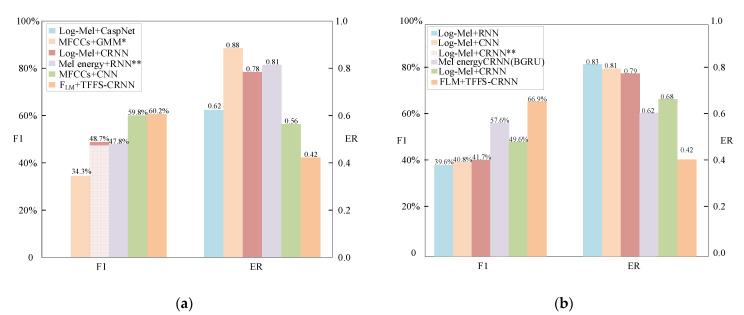
The results of different methods in the evaluation dataset. The figure (**a**) is the experimental result of the DCASE 2016 Task3 dataset; the figure (**b**) is the experimental result of the DCASE 2017 Task3 dataset.

**Table 1 sensors-22-06818-t001:** Event instances per class in DCASE 2016 Task 3 dataset and DCASE 2017 Task 3 Dataset.

DCASE 2016 Task3 Dataset				DCASE 2017 Task3 Dataset	
Residential Area		Home		Street	
Event Label	Instances	Event Label	Instances	Event Label	Instances
(object) banging	23	(object) rustling	60	brakes squeaking	24
bird singing	271	(object) snapping	57	car	110
car passing by	108	cupboard	40	children	19
children shouting	31	cutlery	76	large vehicle	24
people speaking	52	dishes	151	people speaking	47
people walking	44	drawer	51	people walking	48
wind blowing	30	glass jingling	36		
		object impact	250		
		people walking	54		
		washing dishes	84		
		water tap running	47		

**Table 2 sensors-22-06818-t002:** The structure of the TFFS-CRNN model. The bottom is the input layer.

Layer Type	Configurations
Output	output shape = (256,6)
Recurrent	hidden unit number = 32
Recurrent	hidden unit number = 32
Merge	mode = ’mul’
Repeat and Reshape	output shape = (256,128,2)
Softmax activation	-
Feedforward	hidden unit number = 256
Reshape	output shape = 256,256
Permute	output shape = 256,128, 2
Max pooling	sub-sampling rate = 2
ReLU activation	-
Convolution	filter number, kernel size = 128, (3,3)
Max pooling	sub-sampling rate = 2
ReLU activation	-
Convolution	filter number, kernel size = 128, (3,3)
Max pooling	sub-sampling rate = 5
ReLU activation	-
Convolution	filter number, kernel size = 128, (3,3)
Merge	mode = ’TF-Attention’
Multiply on the T/F direction	mode = ’T-Attention’ and ‘F-Attention’
Softmax activation	-
Convolution	filter number, kernel size = 1, (1,1)
ReLU activation	-
Convolution	filter number, kernel size = 32, F(1,3) × 254/T(2,1) × 39
Input	Input shape = (256,40)

**Table 3 sensors-22-06818-t003:** The results of different features on TFFS-CRNN.

DCASE 2016 Task3			DCASE 2017 Task3		
Feature	F1	ER	Feature	F1	ER
Log-Mel	58.1%	0.42	Log-Mel	63.2%	0.51
MFCCS	57.8%	0.45	MFCCS	58.5%	0.55
STFT	50.9%	0.52	STFT	55.1%	0.60
*F_MS_*	58.5%	0.43	*F_MS_*	63.7%	0.52
*F_LS_*	59.0%	0.46	*F_LS_*	63.2%	0.57
** *F_LM_* **	**60.2%**	**0.40**	** *F_LM_* **	**66.9%**	**0.42**

**Table 4 sensors-22-06818-t004:** The results of fusion strategies on TFFS-CRNN.

DCASE 2016 Task3			DCASE 2017 Task3		
**Strategy**	**F1**	**ER**	**Strategy**	**F1**	**ER**
LM-T&F-Average	58.6%	0.42	LM-T&F-Average	61.4%	0.46
**LM-T&F-Weight**	**60.2%**	**0.40**	**LM-T&F-Weight**	**66.9%**	**0.42**
LM-T&F-Channel	55.3%	0.47	LM-T&F-Channel	58.9%	0.55
L-T&F-Average	52.6%	0.49	L-T&F-Average	57.4%	0.45
**L-T&F-Weight**	**55.1%**	**0.41**	**L-T&F-Weight**	**59.1%**	**0.46**
L-T&F-Channel	50.3%	0.48	L-T&F-Channel	56.9%	0.50

**Table 5 sensors-22-06818-t005:** The results of different classifiers.

DCASE 2016 Task3			DCASE 2017 Task3		
Classifiers	F1	ER	Classifiers	F1	ER
DNN	36.0%	0.76	DNN	41.3%	0.90
CNN	34.8%	0.52	CNN	43.4%	0.77
RNN	32.7%	0.65	RNN	38.6%	0.81
CRNN	38.3%	0.56	CRNN	43.7%	0.75
FS-CRNN	41.6%	0.49	FS-CRNN	48.2%	0.57
TF-CRNN	55.1%	0.41	TF-CRNN	59.1%	0.46
**TFFS-CRNN**	**60.2%**	**0.40**	**TFFS-CRNN**	**66.9%**	**0.42**

**Table 6 sensors-22-06818-t006:** The results of different methods.

DCASE 2016 Task3			DCASE 2017 Task3		
Methods	F1	ER	Methods	F1	ER
MFCCs+GMM [42] *	34.3%	0.88	Log-Mel+RNN [15]	39.6%	0.83
Binaural Mel energy +RNN [45] **	47.8%	0.81	Log-Mel+CNN [16]	40.8%	0.81
MFCCs+CNN [44]	59.8%	0.56	Log-Mel+CRNN [17] **	41.7%	0.79
Log-Mel+CaspNet [45]	-	0.62	Binaural Mel energy+ CRNN(BGRU) [47]	57.6%	0.62
Log-Mel+CRNN [46]	48.7%	0.78	Log-Mel+CRNN [48]	49.6%	0.68
**F_LM_+TFFS-CRNN**	**60.2%**	**0.40**	**F_LM_+TFFS-CRNN**	**66.9%**	**0.42**

The models with “*” are the baseline system of DCASE challenge. The models with “**” are the winning system of DCASE challenge.

## Data Availability

Publicly datasets DCASE 2016 Task3 and DCASE 2017 Task3 were available in this paper. The dataset can be found on https://dcase.community/challenge2017/index and https://dcase.community/challenge2016/index (accessed on 2 September 2022).

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
