# Peer review of "Polyphonic Sound Event Detection Using Temporal-Frequency Attention and Feature Space Attention"

_sensors, 2022, doi:10.3390/s22186818_

Round 1
Reviewer 1 Report
-
The manuscript entitled `Polyphonic Sound Event Detection using Temporal-Frequency Attention and Feature Space Attention` is clearly written well organized and structured, but I have major concerns in the manuscript, therefore my recommendation is the major revision with resubmission after incorporating the below points.
-
The datasets used in the manuscript are not clearly explained, it would be better to use the table to reveal the whole information about the datasets.
-
Authors used very few performance evaluation metrics like error and F1, other important metrics are missed like i.e. Accuracy, precision, recall, etc. also add them in your results for better comparison.
-
Why authors need to convert the MEL and MFCC spectrograms into further harmonic and percusive spectrograms? Also authors did not explain the details of this conversion method.
-
The structure and parametric explanation of the implemented CNN + Feature spaceattention+RNN+FC layer by layer not explained by the authors. The model summary can be helpful in this regard. Only Figure 5 is not adequate, table should also be added.
-
The computational cost (training and testing time ) of the proposed methodology is also missed in the manuscript, thus the proper comparison with other methodologies can not be possible.
-
In equation 9 and 10 the two parameters Alpha and Beta are used, why authors used these parameters, whats the basic purpose or significant role of using these parameters, not clearly explained by the authors.
-
Why authors uses the 1*3 kernel size for frequency attention and 2*1 for temporal attention? Is there any specific reason or it is randomly selected?
-
One of the major concern is authors compare the proposed methodology (2022) with the already existed research from 2013, 2016, 2017, authors should compare with some of the recent studies on the identical datasets.
-
Author Response
Please see the attachment. Thank you very much for your guidance.

Reviewer 2 Report
This paper proposes a polyphonic sound detection scheme. The description of the system is clear and the evaluation results show that proposed system provides better results than other previously proposed algorithms. However, there are some issued that must be attended.
1. The characteristics of the CNN used shown be explained.
2. Please explain how the parameters alpha and beta in equations (9) and (10) are determined.
3. It should be convenient to include a diagram of the blocks included in figure 4.
4. Some fonts of figure 5 -8 are too small and then should be increased, even inside the same figure or including them in the caption.
Author Response

(The authors gave the same response as above.)

Round 2
Reviewer 1 Report
Authors have successfully incorporate all the mentioned points, this manuscript can be accepted in modified version
Reviewer 2 Report
The authors have attended the reviewers comments and improved it according to them.